# Clinical Theragnostic Relationship between Drug-Resistance Specific miRNA Expressions, Chemotherapeutic Resistance, and Sensitivity in Breast Cancer: A Systematic Review and Meta-Analysis

**DOI:** 10.3390/cells8101250

**Published:** 2019-10-14

**Authors:** Rama Jayaraj, Madurantakam Royam Madhav, Sankaranarayanan Gomathi Nayagam, Ananya Kar, Shubhangi Sathyakumar, Hina Mohammed, Maria Smiti, Shanthi Sabarimurugan, Chellan Kumarasamy, T. Priyadharshini, K. M. Gothandam, N Ramesh, Ajay Gupta, Siddhartha Baxi, Suja Swamiappan, Sunil Krishnan

**Affiliations:** 1College of Health and Human Sciences, Charles Darwin University, Darwin 0810, Australia; gothandam@gmail.com (K.M.G.); drpnramesh@gmail.com (N.R.); 2Vellore Institute of Technology (VIT), School of Bio-Sciences and Technology, Vellore 632014, India; madhav.sridaran@gmail.com (M.R.M.); carthysgn@gmail.com (S.G.N.); anukar97@gmail.com (A.K.); shubhangi.sathyakumar2015@vit.ac.in (S.S.); hinam606@gmail.com (H.M.); maria.smiti2015@vit.ac.in (M.S.); 3CHIRI, School of Pharmacy and Biomedical Research, Faculty of Health Sciences, Curtin University, Bently campus, Western Australia; drshanthisg@gmail.com; 4North Terrace Campus, University of Adelaide, Adelaide 5005, Australia; chellank54@gmail.com; 5Department of Biochemistry, Bharathiyar University, Coimbatore, Tamil Nadu 641046, India; santhpriya8@gmail.com (T.P.); sujaramalingam08@gmail.com (S.S.); 6National Heart Institute, New Delhi 110065, India; oncoldr@gmail.com; 7John Flynn Private Hospital, Genesis Cancer Care, Tugun 4224, Australia; Siddhartha.Baxi@genesiscancercare.com; 8Department of Radiation Oncology, Division of Radiation Oncology, The University of Texas MD Anderson Cancer Center, Houston, TX 77030, USA; SKrishnan@mdanderson.org

**Keywords:** miRNAs, chemoresistance, breast cancer, systematic review, meta-analysis

## Abstract

Awareness of breast cancer has been increasing due to early detection, but the advanced disease has limited treatment options. There has been growing evidence on the role of miRNAs involved in regulating the resistance in several cancers. We performed a comprehensive systematic review and meta-analysis on the role of miRNAs in influencing the chemoresistance and sensitivity of breast cancer. A bibliographic search was performed in PubMed and Science Direct based on the search strategy, and studies published until December 2018 were retrieved. The eligible studies were included based on the selection criteria, and a detailed systematic review and meta-analysis were performed based on PRISMA guidelines. A random-effects model was utilised to evaluate the combined effect size of the obtained hazard ratio and 95% confidence intervals from the eligible studies. Publication bias was assessed with Cochran’s Q test, I^2^ statistic, Orwin and Classic fail-safe N test, Begg and Mazumdar rank correlation test, Duval and Tweedie trim and fill calculation and the Egger’s bias indicator. A total of 4584 potential studies were screened. Of these, 85 articles were eligible for our systematic review and meta-analysis. In the 85 studies, 188 different miRNAs were studied, of which 96 were upregulated, 87 were downregulated and 5 were not involved in regulation. Overall, 24 drugs were used for treatment, with doxorubicin being prominently reported in 15 studies followed by Paclitaxel in 11 studies, and 5 drugs were used in combinations. We found only two significant HR values from the studies (miR-125b and miR-4443) and our meta-analysis results yielded a combined HR value of 0.748 with a 95% confidence interval of 0.508–1.100; *p-value* of 0.140. In conclusion, our results suggest there are different miRNAs involved in the regulation of chemoresistance through diverse drug genetic targets. These biomarkers play a crucial role in guiding the effective diagnostic and prognostic efficiency of breast cancer. The screening of miRNAs as a theragnostic biomarker must be brought into regular practice for all diseases. We anticipate that our study serves as a reference in framing future studies and clinical trials for utilising miRNAs and their respective drug targets.

## 1. Introduction

Breast cancer is the most prevalent type of cancer in women worldwide [1]. This makes it a cause of increasing concern, and it is important to address this issue. It was estimated that 41,070 breast cancer deaths occurred in women during 2017 in the USA alone, making it the second-leading cause of cancer-related death in women [1]. A large number of breast cancer patients are from developing countries as compared to Western countries, mainly due to their increasing populations [2]. In developed countries, breast cancer is often diagnosed early and treated accordingly; developing countries have higher death rates due to delayed diagnosis and improper access to healthcare [2]. Regardless of this, in developed countries breast cancer is second to lung cancer for cancer-related deaths in women [2]. Asia has 44% of the world’s breast cancer deaths, with 39% of overall new breast cancer cases diagnosed [2]. In India, breast cancer has been ranked as the foremost cancer among the Indian female population [3]. Approximately 25% of female cancer cases in the country are breast cancer [4,5]. The rate of incidence was found to be 25.8 in 100,000 women, and the mortality rate was 12.7 per 100,000 women (2017 statistics) [3]. The highest rate of incidence was found to be in Delhi (41 per 100,000 women) followed by Chennai (37.9 per 100,000 women), Bangalore (34.4 per 100,000 women) and Thiruvananthapuram district (33.7 per 100,000 women) [3]. When the mortality-to-incidence ratio was analysed, it was found to reach 66 in rural registries and 8 in urban registries [3]. Another troubling concern about the scenario of breast cancer in India is the increased incidence of disease in younger Indian women (between the ages of 30 and 40) [3,4,5]. Presently, almost 48% of breast cancer patients in India are below 50 years of age [4,5]. There is an increasing trend of breast cancer in women between the ages of 25 to 40 in the past 25 years [4,5].

At present, breast cancer is classified into four types: (1) Luminal A (classical hormone-positive tumours); (2) Luminal B (hormone-positive with higher ki 67 and poorer prognosis); (3) Triple-negative (ER/PR/HER neu negative); and (4) Her 2 neu overexpressing [6,7]. Currently, several treatments are available for breast cancer, and these include: surgical resection [8], which is often followed by radiotherapy [9], hormone replacement therapy (differs in pre-menopausal and post-menopausal women) [10], targeted therapies [11], immunotherapy [12] and chemotherapeutic drugs [13]. There are a number of chemotherapeutic drugs that are commonly in use and have distinct mechanisms of action, such as anthracyclines (e.g., doxorubicin [14] and epirubicin [15]), taxanes (e.g., Paclitaxel [16,17], docetaxel [16]), alkylating agents (e.g., cyclophosphamide (CTX) [18], carboplatin [17]), trastuzumab—a monoclonal antibody targeted against Her 2 neu [17], anti-metabolites (e.g., 5-fluorouracil (5-FU)) [18], and hormonal agents (e.g., tamoxifen, estradiol (E2), fulvestrant, anastrazole, letrozole).

Conventional chemotherapeutics for breast cancer treatment comprise cytotoxic [19], hormonal [20], and immunotherapeutic agents [21]. Both in neoadjuvant and adjuvant instances, the effectiveness of the chemotherapeutics is limited by resistance developed in the tumour tissue. This is mainly due to the various genetic and epigenetic changes found in cancer cells, and the resistance thus conferred may be intrinsic or acquired [22]. Like most other tumour cells, breast cancer cells exhibit the phenomenon of multi-drug resistance (MDR) [23]. MDR is characterized by a combination of mechanisms including, P-glycoprotein (P-gp) [20], multidrug-resistance-associated protein 1 (MRP1) and breast cancer resistance protein (BCRP) of the ATP-binding cassette (ABC) membrane transporter family, which efflux a diverse range of anticancer drugs from the tumour cells [23,24]. Other notable mechanisms that simultaneously contribute to MDR are enhanced aldehyde dehydrogenase (ALDH) activity, up-regulation of anti-apoptotic B-cell lymphoma-2 (Bcl-2) family proteins and abnormal activation of signalling pathways such as PI3K (phosphatidylinositol 3-kinase)/Akt, Notch, Hedgehog and Wnt pathways [25,26,27]. These mechanisms are predominantly showcased in CSCs (cancer stem cells).

The recent surge in the number of cancer cases along with the development of drug resistance in a large number of tumours has pushed the direction of cancer research towards new arenas that provide the grounds for the development of more effective personalised medicine treatment. MicroRNAs (miRNAs) pave the way for this by being potential biomarkers for early cancer detection, and could also help in designing a more specific treatment plan by helping in the analysis of drug resistance and sensitivity [28]. Various studies have been conducted highlighting the effect of miRNAs in chemotherapeutic resistance in cancers such as gastric cancer [29], breast cancer [30], cervical cancer [31], colorectal cancer [32], lung cancer [33], oral cancer [34], ovarian cancer [35], pancreatic cancer [36], prostate cancer [37] and skin cancer [38].

In one study it was found that there was increased resistance to docetaxel in breast cancer tissues having decreased expression of miR-638, and the restoration of miR-638 in these tissues led to apoptosis and enhanced sensitivity to docetaxel [39]. Microarray miRNA expression analysis in OHT (4-hydroxytamoxifen) showed the overexpression of eight miRNA genes, namely, miR-221, miR-222, miR-181, miR-203, miR-375, miR-32, miR-171, and miR-213, as compared to regular MCF-7 cell line conferring resistance [40]. Furthermore, seven miRNAs were under-expressed in OHT cells: miR-342, miR-484, miR-21, miR-24, miR-27, miR-23 and miR-200. miR-221 and miR-222 were also found to be up-regulated in HER2/neu-positive primary human breast cancer cells [40].

When an MCF7 (Michigan Cancer Foundation-7 cell line treated with VP-16 (etoposide) was compared with the untreated parent MCF7 cell line, it was observed that 17 miRNAs had abnormal levels of expression; the majority of them were up-regulated, whereas miR-326, miR-429, miR-187, miR-7, and miR-92-2 showed decreased expression [41]. The results were verified by RT-PCR, and it was concluded that these miRNAs could be specific regulators of MRP1 (multidrug-resistance-associated protein) and play a critical role in MDR (multiple drug resistance) [41].

A clinical study comparing the effects of the drug tamoxifen versus tamoxifen plus breast radiotherapy, carried out on 71 lymph-node-negative (LNN) breast cancer patients, revealed that the up-regulation of miRNA-301 in co-operation with SKA2 (spindle kinetochore-associated complex subunit 2) increased proliferation, migration, invasion and tumour formation through the regulation of key signalling pathways including PTEN, FOXF2 and Col2A1 [42]. According to another study, high levels of miRNA-210 expression in plasma was observed to be associated with trastuzumab resistance in HER-2 (human epidermal growth factor receptor 2)-positive breast cancer patients [43]. Xiang Ao and his colleagues examined 55 pairs of breast cancer tissues and adjacent normal tissues in total, and found that resistance to taxol in breast cancer patients increased with the loss of miRNA-17 and miRNA-20b, by the up-regulation of nuclear receptor co-activator 3 (NCOA3) levels [44].

Over the years, several studies have focused on the role of various miRNAs in the chemotherapeutic resistance or sensitivity in breast cancer. However, none of these studies have been able to conclusively define the exact mechanism by which these miRNAs are involved in chemo-sensitivity/resistance. Through this study, we aim to provide insight into the association of the expression of specific miRNAs with breast-cancer-related chemotherapeutic drug resistance and sensitivity, thereby making it relevant in a clinical setting. Further, this study paves the way to devise new treatment strategies targeting these miRNAs, and developing alternate ways to counter the occurrence of chemo-resistance in breast cancer. This study was carried out with the aid of tools including meta-analysis and systematic review.

## 2. Methods

To obtain studies to perform the meta-analysis, two databases were extensively used: PubMed and Science Direct. This systematic review required articles related to the chemotherapeutic resistance specific to miRNA in breast cancer. To obtain relevant papers, the selection was performed using of the following MeSH (Medical Subject Heading) terms: “miRNA” or “microRNA”, “drug resistance” and “breast cancer”. To further refine the process of selection, only papers published within 2012–2018 were selected. This systematic review and meta-analysis study adheres to the Preferred Reporting Items for Systematic Review and Meta-Analysis (PRISMA) guidelines [45].

The study search ended on (31 December 2018). After the initial screening process, additional studies were obtained via the reference section of relevant articles. The relevance of articles was determined by reading the title and abstract followed by the analysis of the complete text. The search was conducted in an orderly and elaborate manner, and was designed to meet the requirements of the study.

### 2.1. Selection Criteria

Studies that were to be used in the systematic review and meta-analysis had to adhere to certain selection criteria. These criteria were of two types: inclusion criteria and exclusion criteria. The inclusion criteria set the guidelines for the studies that could be included in the analysis process and included the following factors:An analysis of the association between miRNA and breast cancer;Studies with both breast cancer patients as well as in vitro studies with cell lines;Studies that focused on cancer tissues that had resistance to some form of therapy;Reporting of miRNA profiling platforms;Information about the genes or pathways involved in chemotherapeutic resistance or sensitivity;Inclusion of some in vitro assays to analyse the expression of miRNA or gene-related studies.

Some studies were not considered because of certain exclusion criteria. These included studies that were not in the English language, did not involve drug resistance in breast cancer, studies involving microbes and those focusing only on long non-coding (lnc) RNA. Additionally, review articles, editorials and studies with only in vitro or only breast cancer patient samples were excluded.

### 2.2. Data Analysis

The studies were evaluated separately by both authors (RJ and MRM), and further elaboration was performed with the help of corresponding authors. All articles were subject to the exclusion and inclusion criteria. An MS Excel worksheet (Master) was used to structurally store all the information obtained from the studies that qualified for final inclusion. After a complete survey of full-text and supplementary material, the data from all the studies were broken down under the following important headings: First author, Year of publication, Patients information, Location of the study, Ethnicity, Gender, Drug used, Clinical stages, Number of samples, Lymph node metastasis, Cell lines used, miRNA(s) involved, miRNA profiling platform and Drug pathways or gene associated. A number of biochemical and molecular assays were used to qualitatively and quantitatively analyse the miRNA expression in various studies. The frequency of their usage in all studies were compared and duly represented in a graphical form.

For further qualification of the studies, they had to pass a set of criteria that ensured a degree of quality control [46,47,48]. Two of the authors (RJ and MRM) critically assessed the quality of eligible articles for epidemiological studies based on some checklists derived from Dutch Cochrane Centre represented by Meta-analysis Of Observational Studies in Epidemiology (MOOSE) [49]. The studies that were finally selected had to meet all the criteria as determined by the authors. This process of sorting all the information obtained was a step that was crucial to ensure efficient examination of the studies.

### 2.3. Publication Bias

On the basis of a few distinct methods, two of the authors (RJ and MRM) individually assessed the risk of bias [50,51,52,53,54]. This included the number of patients, year of publication, study period, study location and diagnostic procedure. With the information obtained from the eligible studies, the reviewers arrived at a decision [55,56,57,58,59]. Egger’s and Begg’s bias indicator tests were employed to infer the publication bias along with the inverted funnel plot [60,61,62,63]. The effect size of statistically non-significant, unpublished and small studies was addressed using classic [64] and Orwin [65] fail-safe N tests. Duval and Tweedie’s trim and fill calculation was also performed to compute the new size effect, after the removal of an extremely positive and small study, until a symmetric funnel plot was obtained [66]. A third reviewer was consulted to resolve any disagreement regarding the decision of the team.

### 2.4. Statistical Analysis

We used the Comprehensive Meta-Analysis (CMA) 3.0 software for the meta-analysis and calculated the hazard ratios (HRs) with 95% confidence intervals (CIs). Cochran’s Q test and Higgins’ I^2^ statistic [67] were used to obtain the heterogeneity, and statistical significance was defined as a *p-*value less than 0.01. A fixed-effect model [67] or random-effects model [68] was used to calculate 95% CI in cases where significant heterogeneity was not observed. The overall standard deviation (SD) of each sample from the main sample was calculated using the statistical Z-test.

## 3. Results

The eligible studies for our systematic review and meta-analysis through search results identified are shown in the form of the flow chart in Figure 1. Of the 4584 potential studies, 600 were screened for further proceeding and 92 articles were analysed in depth. Finally, 85 studies were found to be confined to the inclusion and exclusion criteria and the eligible studies involved 5159 tissues. The main characteristics of the patients are represented in Table 1. The systematically reviewed articles met all the criteria, and of the 85 articles included only 6 had hazard ratios and 95% confidence intervals and among these 3 articles denoted them directly in the article and 3 were extracted from Kaplan–Meier curves through online software. Between the 85 articles published, 57 were from China, 9 were from the USA, 5 were from Japan, 3 were from India, 2 each were from France, Italy and Taiwan, and there was 1 from each of Argentina, Canada, Finland, South Korea and Spain. Thirty studies used frozen tissues samples, 15 studies used formalin fixed paraffin embedded (FFPE) samples, 6 studies used core needle biopsy and 1 used blood sample. Meanwhile, 33 studies did not mention the type of material used.

A total of 22 cell lines were used in the 85 studies, and MCF-7, SKBR3, T47D and MDA-MB-231 cell lines were the most frequently included, with MCF-7 used in 33 studies. Zhao Y et al. (2011) used the highest number of cell lines in a single study [135].

Overall, 188 miRNAs were studied in our systematic review and meta-analysis, conjointly 96 miRNAs were upregulated and 87 miRNAs were downregulated. Elevated expression of miR- 18a, 21, 21-3p, 29a, 31, 34a, 34c-5p, 124, 125b, 130b, 137, 138, 138-5p, 139, 139-5p, 140, 140-3p, 141, 149, 149-3p, 155-5p, 181a-5p, 181b, 181b-5p, 181d, 183-5p, 197, 197-3p, 200a-5p, 200c, 205, 210, 210-3p, 221, 222, 378a-3p, 423, 423-5p, 520h, 574, 574-3p, 663, 671, 671-5p, 744, 744-5p, 944, 1246, 1268a, 3178, 3613, 3613-5p, 4258, 4298, 4438, 4443, 4644, 6780b, 6780b-3p, 7107, 7107-5p, 7847, 7847-3p, Let-7a and Lin28 and redundant expression of miR-7, 10b-5p, 17, 20a, 20b, 21, 24-2, 25, 25-3p, 27b, 31-5p, 34a-3p, 103, 125a-3p, 125, 125b-5p, 128, 134, 145, 148a, 149, 181a, 191, 195, 195-5p, 200c, 210, 221, 222, 301a, 320a, 375, 424, 451, 489, 520b-5p, 532-3p, 548n, 574-3p, 708-3p, 873 and Let7a were associated with chemotherapeutic resistance and increased expression of miR- 16, 27a, 34a, 128, 148a, 152, 155, 210, 221, 346, 484 and Let-7 and reduced expression of miR- 21, 24, 23b, 26a, 26b, 27b, 27b-3p, 34a, 100, 125a-3p, 125b-1, 130a-3p, 139, 145, 181a, 181b, 195, 200, 200c, 205, 214, 216b, 218, 301, 320a, 326, 342, 370, 378a-3p, 451a, 489, 576-3p, 638, 760, 765, 1254, Let-7 and Let-7a were associated with chemosensitivity.

Five miRNAs were differentially regulated and four miRNAs (i.e., miR- 90b, 130a, 200b and 452) contributed to chemoresistance. miR-491-3p did not have any impact on chemoresistance or sensitivity. Chemotherapeutic resistance and chemosensitivity were boosted by the miRNAs through drug-regulated cellular pathways. In total, 26 drugs were studied in the included articles: 5-FU, anastrozole, cisplatin, cyclophosphamide, docetaxel, doxorubicin, E2, epirubicin, etoposide, fulvestrant, gemcitabine, lapatinib, letrozole, methotrexate, mitoxantrone, Paclitaxel, PiB, tamoxifen, topotecan, trastuzumab, vinorelbine and combinations such as cisplatin plus doxorubicin, epirubicin plus Paclitaxel, Paclitaxel plus carboplatin, taxol plus doxorubicin plus Cyclophosphamide, Methotrexate, Fluorouracil (CMF), and anthracycline plus taxane were studied, and radiotherapy was also observed in one study. 

### miRNA Pathway Relation

The miRNA and pathways involved in chemoresistance and chemosensitivity are represented in Table 2 and Table 3, respectively.

The relationship between miRNA expression and patient survival was assessed by meta-analysis. Breast cancer (BC) patients had elevated expressions of miR-125b (HR = 6.350, 95% CI = 1.211–33.297), 484 (HR = 0.375, 95% CI = 0.193–0.730), 520h (HR = 1.233, 95% CI = 0.890–1.707), 4443 (HR = 0.721, 95% CI = 0.529–0.983) and downregulated expression of miR-200c (HR = 0.433, 95% CI = 0.102–1.829), 489 (HR = 0.703, 95% CI = 0.415–1.191). An extensive examination found that 89 out of 95 articles did not mention the HR and 95% confidence interval values and of the six remaining articles, only three mentioned them in their manuscript and three HR values were obtained from Kaplan–Meier curve through online software. So, 89 studies were excluded from our meta-analysis due to insufficient data. Cumulatively, a meta-analysis was done for six studies encompassing 852 samples (Figure 2).

An unbiased correlation was observed from Begg and Mazumdar rank collection test results. Regarding Duval and Tweedie’s trim and fill calculation for the fixed-effect model, the point estimate and 95% confidence interval for the combined studies was 0.83921 (0.69115–1.01899). Under the random-effects model, the point estimate and 95% confidence interval for the combined studies was 0.79909 (0.50575–1.26256). Using trim and fill, these values were unchanged. Egger’s regression intercepted at −0.132 with 95% CI from −5.141 to 4.877; t = 0.07, *p* = 0.945. The 1-tailed *p*-value was 0.47237, and the 2-tailed *p*-value was 0.94473. The funnel plot is represented in Figure 3.

## 4. Discussion

This systematic meta-analysis of “the miRNAs that influence the chemoresistance or chemosensitivity to drugs in breast cancer” carefully reviewed over 400 research articles through a systematic PubMed search query from which 80 research articles were scrutinized based on the inclusion criteria.

From the meta-analysis, the results indicate that many miRNAs could intricately orchestrate cellular functions including chemosensitivity/resistance through post-transcriptional control on target gene expression, either canonically or non-canonically. Of the studies included in this meta-analysis, anthracyclines like doxorubicin and epirubicin were predominantly tested in patients/cell lines to study the differential expression of miRNAs followed by tamoxifen in the case of Estrogen Receptor (ER) positive subjects and trastuzumab in the case of Human Epidermal growth factor Receptor (HER) positive subjects. A major limitation in our research is that less than 10% of the 80 papers (6 papers) had direct hazard values that could be utilized for the meta-analysis, reducing the accuracy of the results obtained since only a small fraction of papers were used to give results of the whole, leading to the biasing of the results. There is a possibility of our interpretation being wrong in the context of heterogeneous disease.

### 4.1. Role of miRNAs in Guiding Diagnosis and Prognosis

We extracted the prognosis results of six miRNAs from six different studies. Among the selected miRNAs, two miRNAs (miR200c and miR489) were downregulated and the remaining four miRNAs (miR484, miR4443, miR520h and miR125b) were upregulated. Both downregulated miRNAs were associated with better prognosis; similarly, both miRNAs (miR484 and miR4443) from the overexpressed miRNAs were expressed as better prognosis whereas miR520h and miR125b were associated with poor prognosis.

The overall hazard ratio (95% CI) of the prognostic significance was 0.78 (0.508–1.100) at a *p*-value of 0.140 which was analysed by random-effect model. This overall combined sized effect estimate indicates that the miRNAs decreased the likelihood of death of breast cancer patients by 22%. This means an HR value >1 indicates an increased risk of breast cancer survival whereas an HR <1 indicates a decreased risk of breast cancer patient survival. The Z-value of the overall effect size was −1.476. The individual overall hazard ratios (95% CIs) of upregulated and downregulated miRNAs were estimated 0.662 (0.403–1.087) and 0.904 (0.487–1.678), respectively. On observing the overall effect size of the individual subgroups, the significant prognosis was associated with a good prognosis, and hence the miRNAs could be considered as better prognostic biomarkers for breast cancer patients.

The Z-value of upregulated and downregulated miRNAs for the null hypothesis test (the mean risk ratio of which is 1.0) were −1.636 and −0.319, respectively. Both the differently expressed miRNA subgroups were associated with lower risk of death in breast cancer patients and hence we cannot accept the null hypothesis that the risk is lower in both differently expressed miRNAs. Similar to our study, two other studies have studied the subgroup analysis of higher and lower expressed miRNAs in meta-analysis studies of the prognosis of melanoma and nasopharyngeal carcinoma patient survival. Those studies demonstrated different risk levels among the subgroups, whereas in our study both subgroups exhibited better prognosis for cancer patients. More studies are required to obtain better prognostic significance of miRNAs in breast cancer patients [147].

### 4.2. Current Challenges

Systematic reviews and meta-analytic studies face a number of challenges when investigating the theragnostic relationship between miRNA and chemotherapeutic response in breast cancer. The primary limiting factor for detailed analysis and clinically applicable insights/results is the scarcity of data. The literature in this specific niche of breast cancer treatment is sparse, with few high-quality studies being available for comparison and analysis. This challenge is exacerbated by the lack of homogeneity between similar studies. The variance in study parameters and the methodology makes assessment difficult by introducing uncertainty in the reliability of the results. Furthermore, a large number of studies have explored this topic via the use of in-vitro models, which cannot be directly applied to clinical theragnostics. The lack of well-documented, large-scale, patient-based clinical studies is a significant challenge faced by this study. Furthermore, the mechanisms of miRNA and chemotherapeutic response are not currently understood in detail, requiring further assessment in the future if meta-analytic studies are to provide conclusions viable for application in the clinical sphere.

The strengths of our paper include its large set of research papers, varied results in terms of miRNAs and pathways that show a change in function in cancerous cells. The result of this exhaustive analysis has provided us with a large number of miRNAs that can be focused on for prognostic or diagnostic purposes. Many miRNAs play a role in regulating many vital cellular pathways, and these regulations are observed to be significantly potentiated or deregulated during treatment with chemotherapeutics. A single miRNA can regulate multiple genes, and this regulation down the cascade can affect many pathways. Many reports have independently observed several genes or pathways as targets of many miRNAs. Of those, the treatment of doxorubicin has been frequently observed to affect the PTEN/Akt and MAPK signalling pathways, and increases chemoresistance (Table 3). In the case of miRNA 21 which is also an oncomiR, treatment with Fulvestrant; Selective Estrogen Receptor Degrader (SERD) or trastuzumab (HER2 antagonist) leads to downregulation, affecting the EMT. Whereas treatment with tamoxifen; Selective Estrogen Receptor Modulators (SERM) downregulates the expression of miRNA-21 via estrogen-dependent functions, leading to chemosensitivity.

In case of miRNAs 221 and 222, the treatment with fulvestrant, doxorubicin or trastuzumab also leads to the downregulation with increased expression of ABC transporters. The treatment with Paclitaxel leads to the downregulation of miRNA 320a with downregulation of TRPC5, NFATC3 and the FTS-1 genes, ultimately causing chemoresistance. miRNA 125b is upregulated when treated with tamoxifen, letrozole, anastrazole or fulvestrant due to its interaction with the Akt/mTOR pathway, leading to chemoresistance. The same pattern is observed when treatment of 5-FU, Paclitaxel and cyclophosphamide is applied, which affects the EMT pathway; or when 5-FU is used, which affects the transcription factor E2F3.

miRNAs Let-7, 181a and 145 are also majorly downregulated when treated with drugs like doxorubicin, tamoxifen, or epirubicin, with increases in chemosensitivity. Thus, myriad miRNAs take centre stage in the search for theragnostic miRNAs indicating drug resistance. However, Our study has tried its best to bridge the gaps, and serves as a benchmark for further clinical studies in personalized treatment research.

## Figures and Tables

**Figure 1 cells-08-01250-f001:**
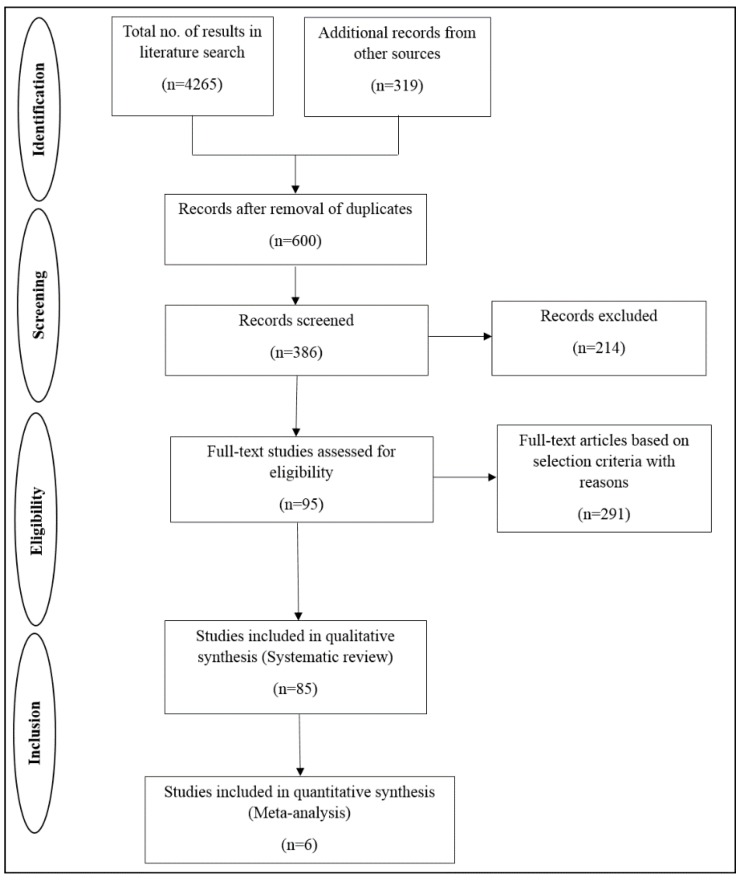
Flowchart of our literature search.

**Figure 2 cells-08-01250-f002:**
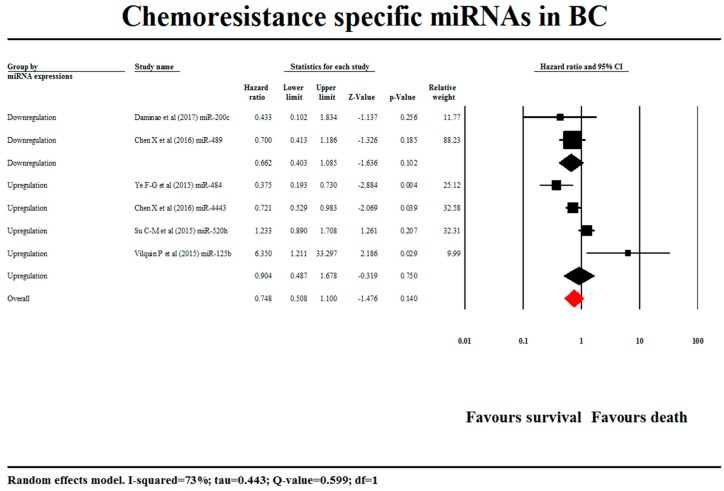
Forest plot of the studies included in our meta-analysis. BC: breast cancer.

**Figure 3 cells-08-01250-f003:**
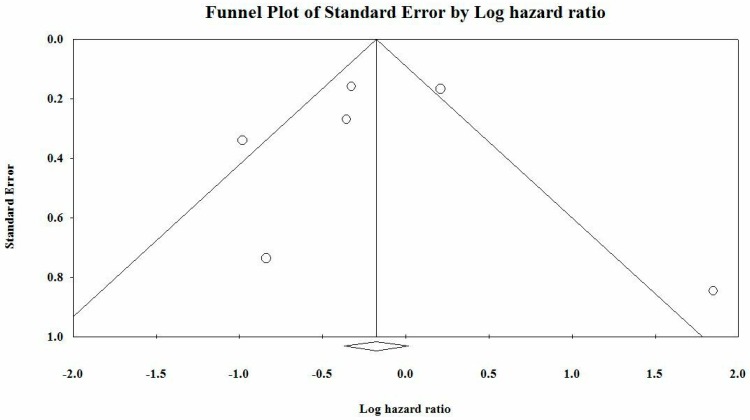
Funnel plot of the studies included in our meta-analysis.

**Table 1 cells-08-01250-t001:** Main characteristics of the included studies.

Author	Ethnicity (Patient)	Period of Study	Drug(s)	Clinical Stages					No. of Samples (Cancer/Normal)	miRNA	miRNA Profiling Platform
				Total stages	I	II	III	IV			
Lin X et al. (2017) [69]	Chinese	2001 to 2006 and 2015	docetaxel	2 stages (I–II and III)	74	4	60	0	138/83	34a	GeneSpring GX (Agilent Technologies, Capital Biochip Corporation)
Zhao G et al. (2017) [39]	Chinese	January 2012 to November 2015	docetaxel	NM	NM	NM	NM	NM	78/78	638	qRT-PCR- SYBR Premix ExTaqTM (Takara, USA)
Nakano M et al. (2017) [70]	Japanese	NM	methotrexate	3 stages (I, I–II, II, II–III)	1	21	1	NM	19/19	25-3p and 125a-3p	Mx3000P (Stratagene, La Jolla, CA)
Miao Y et al. (2017) [71]	Chinese	January 2014 to March 2016	doxorubicin	NM	NM	NM	NM	NM	29/29	130b	SYBR Green qRT-PCR master mix (TaKaRa, Otsu, Shiga, Japan)
Chen M-J et al. (2017) [72]	Taiwanese	NM	tamoxifen	NM	NM	NM	NM	NM	36^a^	148a, 152	ABI 7900 and SYBR® Select Master Mix (Applied Biosystems).
Yang F et al. (2017) [73]	Chinese	2012–2015	docetaxel	NM	NM	NM	NM	NM	24/24	346	ABI 7300 real-time PCR machine (Applied Biosystems, USA)
Gong J-P et al. (2016) [74]	Chinese	July 2010 to June 2014	Paclitaxel	NM	NM	NM	NM	NM	40^a^	24	TaqMan™ MicroRNA Assays (Applied Biosystems; Thermo Fisher Scientific, Inc.)
Ao X et al. (2016) [44]	Chinese	2009–2011	taxol	3 stages (II, III and III-IV)	0	12	18	25	55/55	17 and 20b	SYBR on the CFX96 system (Bio-Rad).
Zhu J et al. (2016) [75]	Chinese	2005–2009	tamoxifen	3 stages (II, III and III–IV)	0	8	22	22	73/19	27b-3p	SYBR on the CFX96 system (Bio-Rad)
Chen X et al. (2016) [76]	Chinese	January 2010 to February 2015	docetaxel, epirubicin and vinorelbine	NM	NM	NM	NM	NM	55/26	29a, 34a, 90b, 130a, 138, 139, 140, 149, 197, 200b, 210, 222, 423, 452, 574, 671, 744, 1246, 1268a, 3178, 3613, 4258, 4298, 4644, 6780b, 7107 and 7847	SYBR® Advantage® qPCR Premix, Light cycler system (Roche, Australia)
Damiano V et al. [77]	Italian	2000–2010	anthracycline, anthracycline + taxane and CMF	2 stages (I–II and III)	2		48	0	51^a^	200c	TaqMan normalizer (Applied Biosystems, ThermoFisher Scientific)
Jana S et al. (2016) [78]	Indian	NM	NM	NM	NM	NM	NM	NM	35/35	216b	SYBR green detection system
Wang D et al. (2016) [79]	Chinese	2010–2015	doxorubicin	NM	NM	NM	NM	NM	21^a^	222	SYBR Premix Ex Taq system (Roche, Australia)
Xu X et al. (2016) [80]	NM	2011–2014	docetaxel	NM	NM	NM	NM	NM	37/37	125a-3p	SYBR Premix ExTaqTM (Takara, USA)
Chen X et al. (2016) [81]	Chinese	January 2010 to February 2015	epirubicin	3 stages (I, II and III)	10	32	4	0	76^a^	4443	MiR-X miRNA qRT-PCR SYBR Kit (638314; Clontech Laboratories, USA)
Gao M et al. (2016) [82]	Chinese	NM	doxorubicin	NM	NM	NM	NM	NM	55/21	145	NCode VILO miRNA cDNA Synthesis Kit and the EXPRESS SYBR GreenER miRNA qRT-PCR Kit, respectively (Invitrogen, Carlsbad, CA, USA)
Thakur S et al. (2016) [83]	Indian	NM	NM	2 stages (I–II and III–IV)	47		38		100/100	21, 145, 195, 210, 221 and Let-7a	TaqMan Universal Master Mix kit (Applied Biosystems, USA)
Hu Y et al. (2016) [84]	Chinese	June 2014 to June 2015	docetaxel, doxorubicin and cyclophosphamide	3 stages (II, III and III–IV)	0	7	19	4	30^a^	205	TaqMan assays (Life Technologies)
Sha L-Y et al. (2016) [85]	Chinese	NM	epirubicin plus Paclitaxel	NM	NM	NM	NM	NM	20/20	18a	TaqMan MicroRNA Assay Kit (Applied Biosystems)
Chen X et al. (2016) [86]	Chinese	2008–2013	doxorubicin	4 stages (I, II, III and IV)	37	64	12	3	114/114	489	SYBR Primescript miRNA RT PCR Kit (TaKaRa, Dalian, China)
Venturutti L et al. (2016) [87]	Argentinians	2008–2014	trastuzumab and lapatinib	4 stages (I, II, III and IV)	5	9	3	2	19^a^	16	TaqMan® MicroRNA assay (Ambion)
Gu X et al. (2016) [88]	Chinese	January 2010 to December 2013	epirubicin and docetaxel	2 stages (II and III)	NM	NM	NM	NM	82/60	451	miScript SYBR Green PCR Kit (QIAGEN, Hilden, Germany) and a real-time LightCycler PCR (Roche Molecular Biochemicals, Mannheim, Germany)
Zhong S et al. (2016) [89]	Chinese	January 2010 to February 2015	docetaxel, epirubicin and vinorelbine	3 stages (I, II and III)	6	8	9	0	23^a^	138-5p, 139-5p, 140-3p, 149-3p, 197-3p, 210-3p, 423-5p, 574-3p, 744-5p, 1246, 1268a, 3178, 4258, 4298, 4443, 4644, 6780b-3p, 7107-5p and 7847-3p	Affymetrix GeneChip miRNA 4.0 Array
Zhang B et al. (2015) [90]	Chinese	NM	Paclitaxel	NM	NM	NM	NM	NM	36/36	100	Realplex Real-time PCR Detection System (Eppendorf, Beijing, China)
Shen R et al. (2015) [91]	Chinese	Between January 2006 to December 2011	tamoxifen	NM	NM	NM	NM	NM	18^a^	155	SYBR Green PCR master mix (TaKaRa) on the ABI 7500HT System
Yu X et al. (2015) [92]	Chinese	NM	tamoxifen and fulvestrant	NM	NM	NM	NM	NM	20/20	214	MiScript SYBR Green PCR kit (Qiagen)
Zhou S et al. (2015) [93]	Chinese	March 2014 to June 2015	cisplatin	NM	NM	NM	NM	NM	40/40	27a	FastStart Universal STBR Green Master (Roche, Switzerland)
Zheng Y et al. (2015) [94]	Chinese	NM	doxorubicin	NM	NM	NM	NM	NM	30/30	181b	TaqMan MicroRNA assays kit (Applied Biosystems, USA)
Ye Z et al. (2015) [95]	Chinese	NM	cisplatin	NM	NM	NM	NM	NM	85/85	221	SYBR Green (Takara)
Mattos-Arruda L-D et al. (2015) [96]	Spaniards	2005–2011	trastuzumab, anthracyclines, taxanes	NM	NM	NM	NM	NM	85^a^	21	LightCycler 480 Real-Time PCR System (Roche)
Lu L et al. (2015) [97]	Chinese	Not mentioned	doxorubicin, cyclophosphamide and fluorouracil	2 stages (II–III)	NM	NM	NM	NM	40^a^	134	SYBR PrimeScript miRNA RT-PCR Kit (Takara, Japan)
Zhang H-d et al. (2015) [98]	Chinese	2012–2015	docetaxel	2 stages (I–II and III)	18		17	0	35^a^	139	TaqMan MicroRNA Assay Kit (assay ID: miR-139-5p: 002289, and RNU6B: 001093), (Applied Biosystems, Life Technologies)
He H et al. (2015) [99]	Chinese	October 2012 to January 2015	cisplatin	NM	NM	NM	NM	NM	70/70	944	ABI PRISM 7900 Sequence Detection System (Applied Biosystems) with SYBR Green (TaKaRa, Japan)
Ikeda K et al. (2015) [100]	Japanese	Not mentioned	tamoxifen	NM	NM	NM	NM	NM	40/16	378a-3p	TaqMan microRNA assays (Applied Biosystems, CA, USA)
Wu J et al. (2015) [101]	Chinese	January 2005 to December 2006	before therapy	NM	NM	NM	NM	NM	39^a^	Let7a	Real-time quantitative reverse transcription PCR (qRT-PCR)
		January 2008 to December 2009	epirubicin	NM	NM	NM	NM	NM	31^a^		
Takahashi R et al. (2015) [102]	Japanese	1996–2000	docetaxel	1 stage (II–III)	NM	26		NM	26/9	27b	TaqMan MicroRNA Assays (Applied Biosystems)
Niu J et al. (2015) [103]	Chinese	1 January 2009 to 31 December 2010	doxorubicin	2 stages (I–II and III–IV)	49		13		62^a^	181a	MyiQ Real-Time PCR Detection System (Bio-Rad)
Su C-M et al. (2015) [104]	Taiwanese	NM	Paclitaxel	2 stages (I and I–II)	36	110	NM	NM	146^a^	520h	Applied Biosystems 7900 Fast Real-Time PCR
Boulbes D et al. (2015) [105]	American	NM	trastuzumab, fluorouracil, epirubicin and cyclophosphamide	NM	NM	NM	NM	NM	50^a^	has-520b-5p, 532-3p, 548n and 34a-3p	miRNA microarray (version 4.0, microRNACHIPv4)
Manvati S et al. (2015) [106]	Indian	NM	docetaxel	3 stages (I, II and III)	NM	NM	NM	NM	46/46	24-2	TaqMan microRNA assays (Applied Biosystems)
Kang L et al. (2015) [107]	Chinese	NM	Paclitaxel	4 stages (I, II, III and IV)	11	18	12	4	45^a^	34a	TaqMan MicroRNA Assay kit (Applied Biosystems, Foster City, CA, USA)
Lu M et al. (2015) [108]	Chinese	2009–2010	tamoxifen	NM	NM	NM	NM	NM	31/27	320a	Applied Biosystems Step One real-time PCR system using an SYBR Premix Ex Taq II Kit (Takara Bio, Inc., Shiga, Japan)
Ye F-G et al. (2015) [109]	Chinese	September 2013	gemcitabine	3 stages (I, II and III)	159		32	NM	400/243	484	SYBR Premix Ex Taq System (TaKaRa)
Vilquin P et al. (2015) [110]	French	NM	letrozole, anastrazole, tamoxifen and fulvestrant	3 stages (I, II and III)	4	18	23	0	65/65	125b	ExiLENT SYBR Green Master Mix and CFX96 (BioRad, Marne-laCoquette, France)
Ujihira T et al. (2015) [111]	Japanese	NM	tamoxifen	NM	NM	NM	NM	NM	19^a^	574-3p	triplicate TaqMan microRNA assays (Applied Biosystems, CA, USA)
Cui J et al. (2014) [112]	Chinese	NM	tamoxifen	NM	NM	NM	NM	NM	NM	873	RNeasy Mini kit (Qiagen, Hilden, Germany) or TRIzol (Invitrogen) reagent. SYBR Green PCR Master Mix reagents using an ABI Prism 7700 Sequence Detection System (Applied Biosystems, Foster City, CA, USA)
Lv J et al. (2014) [113]	Chinese	2008–2009	doxorubicin	NM	NM	NM	NM	NM	NM	31, 125b-1, 141, 145, 196b, 200a, 200c, 370, 429, 491-3p, 576, 760, 765 and Let-7a	ABI 7900 PCR System (Applied Biosystems, USA) using Power SYBR Green PCR Master Mix (2X, Applied Biosystems)
He X et al. (2014) [114]	Chinese	NM	cisplatin	4 stages (I, II, III and IV)	15 + 17	15 + 17	30 + 23	30 + 23	85^a^	218	TRIzol reagent (Invitrogen) miRNA microarray chip (v.10.0, Exiqon, Vedbaek, Denmark)
Winsel S et al. (2014) [115]	Norwegians	May 1995 to December 1998	taxol	NM	NM	NM	NM	NM	101^a^	378a-3p	RNeasy Mini Kit (Qiagen) TaqMan Universal Master Mix II, no PNG (Applied Biosystems, Foster City, CA, USA)
Hu J et al. (2014) [116]	Chinese	NM	NM	4 stages (I, II, III and IV)	20	25	31	4	119^a^	93	TRIzol Reagent (Invitrogen) and the miRNeasy Mini Kit (QIAGEN)
He DX et al. (2014) [117]	Chinese	NM	doxorubicin, Paclitaxel	NM	NM	NM	NM	NM	NM	320a	All-in-One miRNA qRT-PCR detection kit (GeneCopoeia, Rockville, MD, USA)
He DX et al. (2014) [118]	Chinese	NM	doxorubicin, Paclitaxel	NM	NM	NM	NM	NM	NM	149	All-in-One miRNA qRT-PCR detection kit (GeneCopoeia, Rockville, MD, USA). Briefly, total RNA was extracted from MCF-7/WT and ADM cells with TRIzol (Invitrogen, Carlsbad, CA, USA)
Ouyang M et al. (2014) [119]	Chinese	2011 (January–October)	doxorubicin	NM	NM	NM	NM	NM	NM	10b-5p, 21-3p, 31-5p, 125b-3p, 130a-3p, 155-5p, 181a-5p, 181b-5p, 183-5p, 195-5p and 451a	Total RNA was harvested using TRIzol (Invitrogen) and miRNAeasy mini kit (QIAGEN). SYBR Premix EX TaqTM II kit (Takara, Dalian, China)
Luo ML et al. (2014) [120]	Chinese	NM	PiB	NM	NM	NM	NM	NM	NM	200	Total RNA was isolated from miRNeasy kit (Qiagen) and reversely transcribed by miScript PCR starter kit
Jiang L et al. (2014) [121]	Chinese	NM	doxorubicin	NM	NM	NM	NM	NM	NM	489	Total RNA was prepared using TRIzol (Beyotime, China) according to the manufacturer’s instructions.
Ye XM et al. (2014) [122]	Chinese	NM	trastuzumab/Herceptin	NM	NM	NM	NM	NM	NM	375	Total RNA was extracted from each cell line using TRIzol reagent (Invitrogen, USA)
Zhu Y et al. (2013) [123]	Chinese	NM	doxorubicin	2 stages (I and II)	34	9	NM	NM	43^a^	181a	Total RNA was extracted from each cell line using TRIzol reagent (Invitrogen, Carlsbad, CA, USA)
Ye X et al. (2014) [122]	Chinese	NM	trastuzumab	NM	NM	NM	NM	NM	NM	221	Total RNA from each cell line was extracted by TRIzol reagent (Invitrogen, USA)
Yang G et al. (2013) [124]	Chinese	NM	doxorubicin	2 stages (I and II)	9	8	NM	NM	30^a^	195	Total cellular RNA from tissues and cultured cells were isolated using a TRIzol Reagent (Invitrogen)
Pichiorri F et al. (2013) [125]	Americans	NM	fulvestrant	NM	NM	NM	NM	NM	183/57	21, 103, 221 and 222	TaqMan PCR kit (Applied Biosystems) and 7900HT Sequence Detection System (Applied Biosystems)
Wang H-J et al. (2013) [126]	Chinese	January 2010 to December 2011	Paclitaxel, 5-FU, epirubicin and cyclophosphamide	NM	NM	NM	NM	NM	19/19	125b	ABI 7900HT system (Applied Biosystems)
Ji S et al. (2013) [127]	Chinese	2007–2009	taxol + doxorubicin + cyclophosphamide	NM	NM	NM	NM	NM	67/67	128	QRT-PCR
Hu H et al. (2013) [128]	Chinese	October 2003 to July 2010	topotecan, etoposide, doxorubicin, docetaxel and cyclophosphamide	NM	NM	NM	NM	NM	39/39	663	Conventional TaqMan PCR (Bio-Rad)
Masuda M et al. (2011) [129]	Japanese	NM	estradiol (E2)	NM	NM	NM	NM	NM	41^a^	7	PCR was performed in ABI7500 Real-Time PCR System (Applied Biosystems, Foster city, CA, USA)
Li X et al. (2012) [130]	Chinese	2008–2010	doxorubicin, cyclophosphamide (CTX) and 5-fluorouracil (5-FU)	1 stage (II)	0	38	0	0	38/38	34a	SYBR Green PCR Master Mix (Applied Biosystems, Foster City, CA, USA)
Lv K et al. (2012) [131]	Chinese	2002–2010	Paclitaxel, vincristine	NM	NM	NM	NM	NM	9/9	Lin28	Real-time PCR was performed using the TaqMan MicroRNA Reverse Transcription Kit and the Fast Real-Time PCR System (Applied Biosystems, Carlsbad, CA, USA)
Wang H et al. (2012) [132]	Chinese	2009–2010	5-FU (5-fluorouracil)	2 stages (II and III)	0	35	21	0	56/10	10b, 34a, 125b and 155	miRNA-specific TaqMan MicroRNA Assays (Applied Biosystems)
Jung E-J et al. (2012) [43]	Americans, Koreans	NM	trastuzumab, Paclitaxel, fluorouracil, cyclophosphamide and epirubicin	3 stages (I, II and III)	33	31	8	0	72/72	21, 29a, 126 and 210	TaqMan MicroRNA Assay kit (Applied Biosystems, Foster City, Calif)
Chen J et al. (2011) [133]	Chinese	2007–2011	doxorubicin	NM	NM	NM	NM	NM	39^a^	200c	Real-time PCR was performed using SYBR Green PCR Master Mix (Applied Biosystems, USA) on the Stepone plus system (Applied Biosystems, USA)
Zhu Y et al. (2011) [134]	Chinese	2004–2011	NM	3 stages (II, III and IV)	NM	44	29	4	77^a^	128	Mature miRNA expression analysis was conducted using a TaqMan MicroRNA Assays (Applied Biosystems)
Zhao Y et al. (2011) [135]	NM	NM	tamoxifen	NM	NM	NM	NM	NM	29/15	Let-7	mirVana miRNA isolation kit (Ambion Inc., Austin, TX, USA) or from FFPE tissues using the miRNeasy FFPE Kit (Qiagen, Valencia, CA, USA)
Gong C et al. (2011) [136]	Chinese	2008–2009	trastuzumab (Herceptin)	NM	NM	NM	NM	NM	32^a^	21	Total RNA was harvested using TRIzol (Invitrogen) and the RNeasy minikit (Qiagen) according to the manufacturer’s instructions.
Shi W et al. (2011) [42]	NM	NM	NM	3 stages (I, II and III)	8	33	30	NM	71^a^	301	Standard TaqMan MicroRNA Assay (Applied Biosystems)
Cittelly D et al. (2010) [137]	Americans	1978–1993	tamoxifen	3 stages (I, II and III)	72	346	322	NM	791^a^	342	miRVANA RNA Isolation System (Ambion)
Liang Z et al. (2010) [41]	Americans	NM	VP-16, mitoxantrone	3 stages (I, III and IV)	5	NM	10 (III and IV)	10 (III and IV)	35^a^	326	Total RNA was extracted from 70% to 85% confluence of MCF-7 and MCF-7/VP cells with TRIzol (Invitrogen, Carlsbad, CA, USA)
Maillot G et al. (2009) [138]	NM	NM	tamoxifen	2 stages (III and IV)	NM	NM	5	10	15^a^	21, 23b, 26a, 26b, 27b, 181a, 181b and 200c	miRNA microarray analysis was performed as described by Castoldi and colleagues
Iorio M et al. (2009) [139]	Italians	NM	NM	NM	NM	NM	NM	NM	NM	205	TaqMan MicroRNA Reverse Transcription kit and TaqMan MicroRNA Assay were used to detect and quantify mature microRNA-205 (Applied Biosystems)
Miller T et al. (2008) [40]	Americans	NM	tamoxifen	NM	NM	NM	NM	NM	76^a^	221 and 222	The miRNA microarray was performed at the Ohio State University Comprehensive Cancer Center Microarray Core Facility
Yu F et al. (2007) [140]	Chinese	NM	epirubicin	NM	NM	NM	NM	NM	25^a^	Let-7	NM
Li G et al. (2016) [141]	Chinese	2001–2002	tamoxifen	NM	NM	NM	NM	NM	57/57	1254	mirVana miRNA isolation kit (Ambion) using stem-loop RT primers and analysed by qPCR (TaqMan, TaKaRa)
Yu S-J et al. (2018) [142]	Chinese	2003–2009	Paclitaxel and carboplatin	2 stages (II and III)	NM	28	44	NM	110/110	200a-5p	7900HT Fast Real-Time PCR System (Applied Biosystems)
Lee J-W et al. (2017) [143]	South Korean	NM	doxorubicin	2 stages (I–II and III–IV)	28	NM	21	NM	50/50	708-3p	High-Capacity cDNA Reverse Transcription Kit (Life Technologies)
Si W et al. (2018) [144]	Chinese	NM	Paclitaxel	3 stages (I, II and III)	15	38	53	0	106/106	20a	SYBR Premix Ex Taq (TaKaRa, RR420A)
Cheng S et al. (2018) [145]	Chinese	NM	cisplatin and doxorubicin	NM	NM	NM	NM	NM	57/31	137	ABI Prism 7900HT thermal cycler (Applied Biosystems, Foster City, CA, USA)
Hu G et al. (2018) [146]	Chinese	August 2013 to December 2015	doxorubicin	NM	NM	NM	NM	NM	30^a^	125b	ABI PRISM 7900 Sequence Detection system (Applied Biosystems)

NM: Not Mentioned; a: only cancer tissue; CMF: Cyclophosphamide, Methotrexate, Fluorouracil.

**Table 2 cells-08-01250-t002:** Pathways involved in chemoresistance.

Downregulated			Upregulated		
Drug	miRNA	Gene/Pathway	Drug	miRNA	Gene/Pathway
5-FU	134	ABCC1	5-FU	125b	EMT
anastrozole	424	Akt/mTOR pathway	5-FU	125b	Transcription factor E2F3
anthracycline	200c	ZEB1	anthracycline	21	IL-6/STAT3/NF-κB/PI3K pathway.
anthracycline + taxane	200c	ZEB1	cisplatin	944	Bcl2/BNIP3
CMF	200c	ZEB1	cisplatin and doxorubicin	137	FSTL1/integrin β3/Wnt
CTX	134	ABCC1	CTX	125b	EMT
docetaxel	451	NM	CTX	663	HSPG2
docetaxel	24-2	YWHAZ, TP53, SMAD3, ESR1 and CREBBP	docetaxel	663	HSPG2
doxorubicin	145	MRP1	doxorubicin	130b	PTEN/PI3K/Akt
doxorubicin	320a	TRPC5, NFATC3 and ETS-1 gene	doxorubicin	222	PTEN/Akt/cyclin-dependent kinase (p27) pathway
doxorubicin	149	GlcNAc-NDST1	doxorubicin	181b	MMP/caspase pathway
doxorubicin	103	NCL	doxorubicin	663	HSPG2
doxorubicin	222	NCL	doxorubicin	31	MAPK signalling pathway, cytokine–cytokine receptor interaction
doxorubicin	134	ABCC1	doxorubicin	141	MAPK signalling pathway, cytokine–cytokine receptor interaction
doxorubicin	181a	STAT3/NF-kB/MSK1	doxorubicin	200c	MAPK signalling pathway, cytokine–cytokine receptor interaction
doxorubicin	10b-5p	PTEN/Akt, MAPK, RhoA, FOXO3 and PDCD4 genes	doxorubicin	181b-5p	PTEN/Akt, MAPK, RhoA, FOXO3 and PDCD4 genes
doxorubicin	125b-3p	PTEN/Akt, MAPK, RhoA, FOXO3 and PDCD4 genes	doxorubicin	183-5p	PTEN/Akt, MAPK, RhoA, FOXO3 and PDCD4 genes
doxorubicin	155-5p	PTEN/Akt, MAPK, RhoA, FOXO3 and PDCD4 genes	doxorubicin	195-5p	PTEN/Akt, MAPK, RhoA, FOXO3 and PDCD4 genes
doxorubicin	181a-5p	PTEN/Akt, MAPK, RhoA, FOXO3 and PDCD4 genes	doxorubicin	21-3p	PTEN/Akt, MAPK, RhoA, FOXO3 and PDCD4 genes
doxorubicin	31-5p	PTEN/Akt, MAPK, RhoA, FOXO3 and PDCD4 genes	E2	124	EGFR
doxorubicin	200c	MDR1 mRNA	E2	29a	EGFR
doxorubicin	708-3p	ZEB1/CDH2/vimentin	E2	21	EGFR
doxorubicin	125b	HAX-1	E2	181d	EGFR
E2	301a	EGFR	E2	34c-5p	EGFR
E2	20a	EGFR	epirubicin	4443	TIMP2
E2	149	EGFR	epirubicin + Paclitaxel	18a	Dicer
E2	17	EGFR	epirubicin	125b	EMT
E2	25	EGFR	etoposide	663	HSPG2
E2	191	EGFR	fulvestrant	125b	Akt/mTOR pathway
E2	27b	EGFR	letrozole	205	Akt/mTOR pathway
E2	148a	EGFR	Paclitaxel	520h	DAPK2
E2	210	EGFR	Paclitaxel	Lin28	p21, RB, cyclin B1, Akt and Let-7 miRNA
E2	7	EGFR	Paclitaxel	125b	EMT
epirubicin	Let7a	H-RAS/HMGA2	Paclitaxel and carboplatin	200a-5p	TP53INP1/YAP1
epirubicin	Let7a	H-RAS/HMGA2	tamoxifen	222	p27Kip1
epirubicin	451	NM	tamoxifen	221	p27Kip1
fulvestrant	21	NCL	taxanes	21	IL-6/STAT3/NF-κB/PI3K pathway
methotrexate	25-3p	ADAR1/DHFR	taxol	378a-3p	Triggered receptor tyrosine kinase–MAP kinase pathway signalling, suppression of Aurora B kinase
methotrexate	125a-3p	ADAR1/DHFR	topotecan	663	HSPG2
Paclitaxel	320a	TRPC5 gene; NFATC3gene; ETS-1 gene	trastuzumab	21	IL-6/STAT3/NF-κB/PI3K pathway
Paclitaxel	149	GlcNAc-NDST1	trastuzumab	221	PTEN
Paclitaxel	20a	MAPK1/c-Myc	trastuzumab	21	PTEN
tamoxifen	574-3p	CLTC	vincristine	Lin28	p21, RB, cyclin B1
tamoxifen	873	CDK3, Erα			
tamoxifen	424	Akt/mTOR pathway			
taxol	17	NCOA3			
taxol	20b	NCOA3			
trastuzumab	221	NCL			
trastuzumab	375	IGF1R			

anthracyclin: epirubicin/doxorubicin; EMT: Epithelial-Mesenchymal Transition.

**Table 3 cells-08-01250-t003:** Pathways involved in chemosensitivity.

Downregulation			Upregulation		
Drug	miRNA	Gene/Pathway	Drug	miRNA	Gene/Pathway
CTX	205	VEGF/FGF2	5-FU	34a	Notch 1
cisplatin	218	BRCA1	CTX	34a	Notch 1
doxorubicin	489	Smad3, EMT	cisplatin	27a	BAK-SMAC/DIABLO-XIAP Pathway
doxorubicin	181a	Bcl-2	cisplatin	221	BIM/Bcl-2/Bax/Bak
docetaxel	34a	C22ORF28	docetaxel	346	SRCIN1
docetaxel	638	STARD10	doxorubicin	196b	MAPK signalling pathway, cytokine–cytokine receptor interaction
docetaxel	125a-3p	BRCA1	doxorubicin	200a	MAPK signalling pathway, cytokine–cytokine receptor interaction
doxorubicin	195	Raf-1	doxorubicin	34a	Notch 1
docetaxel	139	Notch 1	doxorubicin	451a	PTEN/Akt, MAPK, RhoA, FOXO3 and PDCD4 genes
docetaxel	27b	ENPP1	doxorubicin	429	MAPK signalling pathway, cytokine–cytokine receptor interaction
docetaxel	205	VEGF/FGF2	gemcitabine	484	CDA/Cyclin-dependent kinase
doxorubicin	145	MAPK signalling pathway, cytokine–cytokine receptor interaction	lapatinib	16	CCNJ/FUBP1
doxorubicin	370	MAPK signalling pathway, cytokine–cytokine receptor interaction	tamoxifen	148a	ALCAM
doxorubicin	576-3p	MAPK signalling pathway, cytokine–cytokine receptor interaction	tamoxifen	152	ALCAM
doxorubicin	760	MAPK signalling pathway, cytokine–cytokine receptor interaction	tamoxifen	Let-7	MAPK/Akt, ER-α36
doxorubicin	765	MAPK signalling pathway, cytokine–cytokine receptor interaction	tamoxifen	155	SOCS6-STAT3 signalling pathway
doxorubicin	125b-1	MAPK signalling pathway, cytokine–cytokine receptor interaction	taxol + doxorubicin + cyclophosphamide	128	Bax
doxorubicin	Let-7a	MAPK signalling pathway, cytokine–cytokine receptor interaction	trastuzumab	16	CCNJ/FUBP1
doxorubicin	130a-3p	PTEN/Akt, MAPK, RhoA, FOXO3 and PDCD4 genes			
doxorubicin	205	VEGF/FGF2			
epirubicin	Let-7	HMGA2			
fulvestrant	214	UCP2/PI3K-Akt-mTOR pathway			
mitoxantrone	326	MRP-1			
Paclitaxel	24	ABCB9			
Paclitaxel	34a	Notch 1			
Paclitaxel	100	mTOR			
PiB	200	Pin1			
tamoxifen	342	Cyclin B1, p53, BRCA1 gene			
tamoxifen	27b-3p	NR5A2/CREB1			
tamoxifen	378a-3p	GOLT1A			
tamoxifen	320a	ARPP-19/ERRᵧ, c-Myc, Cyclin D1			
tamoxifen	21	Estrogen-dependent cellular functions			
tamoxifen	181a	Estrogen-dependent cellular functions			
tamoxifen	181b	Estrogen-dependent cellular functions			
tamoxifen	200c	Estrogen-dependent cellular functions			
tamoxifen	23b	Estrogen-dependent cellular functions			
tamoxifen	26a	Estrogen-dependent cellular functions			
tamoxifen	26b	Estrogen-dependent cellular functions			
tamoxifen	27b	Estrogen-dependent cellular functions			
tamoxifen	1254	CCAR1			
tamoxifen	214	UCP2/PI3K-Akt-mTOR pathway			
VP-16	326	MRP-1			

Anthracyclin: epirubicin/doxorubicin.

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
