# Peer review of "Clinical Theragnostic Relationship between Drug-Resistance Specific miRNA Expressions, Chemotherapeutic Resistance, and Sensitivity in Breast Cancer: A Systematic Review and Meta-Analysis"

_cells, 2019, doi:10.3390/cells8101250_

Round 1
Reviewer 1 Report
The authors performed a systematic review and meta-analysis on the role of miRNA in influencing chemoresistance and sensitivity of breast cancer. From the 85 studies, they found that 96 miRNAs were upregulated and 87 were downregulated. The authors concluded that differently expressed miRNAs play a role in guiding the effective diagnostic and prognostic efficiency of breast cancer.
Line133-135: the authors state that “this study paves the way to devise new strategies targeting these miRNA…”, however, the discussion session is vague and does not draw any conclusion. Can the authors discuss about how these miRNAs can play a role in guiding diagnose and prognose? Which miRNAs will have great potential for diagnosis and prognosis?
26 drugs have been studied in the included articles, is there a correlation between drug and miRNAs?
Line 238: The authors mentioned that four miRNAs contribute to resistance. Can the author explain this sentence better? How did the authors come to this conclusion?
Table 1: Can the authors add reference number for each line on the table?
Table 1: description of the table will be easier if appears on the top.
Author Response
To
Ms Lena Ilic
Assistant Editor
Cells (MDPI)
Special issue: MicroRNA as Biomarker
Ms Lena Ilic
We want to thank once again the Cells (MDPI), Editorial and Reviewer’s team for reviewing our manuscript and providing valuable comments. Our team firmly believe that the comments and feedback from your esteemed reviewers were constructive and would enhance the quality of the paper. Please find the revised manuscript to address issues identified by editors and reviewers team in considering the above review paper for publication.
We have amended the changes and highlighted the sections in the manuscript with track changes.
Reviewer 1
Our sincere thanks for your esteemed corrections and valuable comments to improve the quality of our manuscript. It is greatly appreciated.
Line133-135: the authors state that “this study paves the way to devise new strategies targeting these miRNA…”, however, the discussion session is vague and does not draw any conclusion. Can the authors discuss about how these miRNAs can play a role in guiding diagnose and prognose? Which miRNAs will have great potential for diagnosis and prognosis?
The authors’ thank the reviewer for this specific comment. We have added a paragraph about the role of miRNAs in diagnosis and prognosis in the discussion part (Line no: 293-320).
Role of miRNA’s in guiding diagnosis and prognosis
Based on our study, we have found that six miRNAs have studied their prognosis results from their respective studies from six different studies. Among the selected miRNAs, two miRNAs (miR200c and miR489) were downregulated and remaining four miRNAs (miR484, miR4443, miR520h and miR125b) were upregulated. Both the downregulated miRNAs were associated with better prognosis similarly all the two miRNAs (miR484 and miR4443) from the overexpressed miRNAs were expressed as better prognosis whereas miR520h and miR125b were associated with poor prognosis.
The overall hazard risk of the prognostic significance whas 0.78 (0.508-1.100) at p-value 0.140 which is analysed by Random effect model. This overall pooled sized effect estimate indicated that the miRNAs decreased the likelihood death of breast cancer patients by 22%. This means the HR value >1 indicates the increased risk of breast cancer survival whereas the HR<1 indicates the decreased risk of breast cancer patients survival. The Z value of the overall effect size was -1.476. The individual overall hazard risk of upregulated and downregulated miRNAs was estimated 0.662(0.403-1.087) and 0.904 (0.487-1.678) respectively. On observing the overall effect size of the individual subgroups, the significant prognosis was associated with a good prognosis, and hence the miRNAs were considered as possible chance to be considered as better prognostic biomarker for breast cancer patients.
The Z-value of upregulated and downregulated miRNAs for the null hypothesis test (the mean risk ratio of which is 1.0) are -1.636 and -0.319 respectively. Both the differently expressed miRNAs subgroup are associated with lower risk of death in breast cancer patients and hence we cannot accept the null hypothesis that the risk is lower in both differently expressed miRNAs. Similar to our study, two other studies have studied the subgroup analysis of higher and lower expressed miRNAs among the total meta-analysis study towards the prognosis of melanoma and Nasopharyngeal carcinoma patients’ survival analysis. Those studies have demonstrated with different risk level among the subgroups whereas our study both the subgroups has exhibited better prognosis to cancer patients. But still more studies are required to know the better prognostic significance of miRNAs in breast cancer patients [147].
26 drugs have been studied in the included articles, is there a correlation between drug and miRNAs?
Yes, we have shown the results representing a correlation between the drug and miRNA relation in Table 2 and 3.
Line 238: The authors mentioned that four miRNAs contribute to resistance. Can the author explain this sentence better? How did the authors come to this conclusion?
This particular finding mentioned in line no: 244 came from the published study (Chen X et al., 2016). The authors of this particular study showed four miRNAs (miRNAs-90b, 130a, 200b and 452) contribute to drug resistance. Their results strengthened the miRNA-mediated drug resistance of breast cancer resulting from these miRNAs in vivo using FFPE breast cancer specimens. In order to analysis the biomarker role of these miRNAs in response to chemotherapy and their predictive role in prognosis, they analyzed expression levels of them in ineffective group compared to effective group after and before neoadjuvant chemo-therapy. We did not come to this conclusion and have compiled their findings in our systematic review and meta-analysis.
Chen, X.; Lu, P.; Wang, D.-d.; Yang, S.-j.; Wu, Y.; Shen, H.-Y.; Zhong, S.-l.; Zhao, J.-h.; Tang, J.-h.J.G. The role of miRNAs in drug resistance and prognosis of breast cancer formalin-fixed paraffin-embedded tissues. 2016, 595, 221-226.
Table 1: Can the authors add reference number for each line on the table?
The authors’ thank the reviewer for the comment. We have added the reference number to each line on the table.
Table 1: description of the table will be easier if appears on the top.
We have provided as per the journals instructions.
Reviewer 2 Report
Manuscript is of importance to the field, giving a comprehensive span of the body of recent global research in the quest for chemoresistance miRNA biomarker identification and validation.
However, the conclusions of this manuscript should also mention the current challenges involved with the potential employment of miRNAs in breast cancer theranostics.
In addition, the entire manuscript should be scrutinised by a professional scientific English proof-reader as there are myriads of grammatical and typographic errors across the manuscript, prior to publication.
Author Response
Reviewer 2
Our sincere thanks for your esteemed corrections and valuable comments to improve the quality of our manuscript. It is greatly appreciated.
However, the conclusions of this manuscript should also mention the current challenges involved with the potential employment of miRNAs in breast cancer theranostics.
In addition, the entire manuscript should be scrutinised by a professional scientific English proof-reader as there are myriads of grammatical and typographic errors across the manuscript, prior to publication.
The authors’ thank the reviewer for the comments. We have added a paragraph about the current challenges in line no: 318-331.
Many Thanks,
Dr Rama Jayaraj
Meta-Analysis team
Dr Rama Jayaraj PhD, GCTLHE, MPH
Senior Lecturer-Clinical Sciences
College of Health and Human Sciences
+61 8 8946 6146 | F. +61 8 8946 7474
rama.jayaraj@cdu.edu.au
www.cdu.edu.au
CHARLES DARWIN UNIVERSITY
Reviewer 3 Report
The authors present an elaborate meta study on the role of miRNA in breast cancer treated with chemotherapy. Altogether 85 studies from more than 4000 were eligible for this study but only 6 studies could be included in the meta-analysis as these were the only studies with data for HR and confidence interval.
Although it is an interesting project to analyze miRNAs in a meta-study, the overall result is somehow disappointing. Consequently the authors state as conclusion that they tried to bridge the gaps and call for further studies.
Nevertheless, it is important work to collect all these studies and publish this list for the scientific audience. However, the manuscript reads more like a review than a meta-study. I think it would make the manuscript more interesting when the pathways tackled by the miRNAs in the last 6 studies could be discussed in more detail.
More detailed points:
I am not sure what the meaning of a “pooled HR” really (see abstract) is, as the miRNAs tested, target different pathways and I doubt it is biologically not really meaningful to pool them together. How was the up and down-regulation handled?The introduction gives many epidemiological data on breast cancer and I do not think that this this really relates to the topic of this study.
Maybe the selection criteria have been too strict. Is it really necessary that a study needs “both breast cancer patients’ as well as in vitro studies with cell-lines”?
Author Response
Our sincere thanks for your esteemed corrections and valuable comments to improve the quality of our manuscript. It is greatly appreciated.
Nevertheless, it is important work to collect all these studies and publish this list for the scientific audience. However, the manuscript reads more like a review than a meta-study. I think it would make the manuscript more interesting when the pathways tackled by the miRNAs in the last 6 studies could be discussed in more detail.
The authors’ thank the reviewer for this valuable comment. We have discussed about the six studies in detail. (Line no: 290-317)
More detailed points:
I am not sure what the meaning of a “pooled HR” really (see abstract) is, as the miRNAs tested, target different pathways and I doubt it is biologically not really meaningful to pool them together. How was the up and down-regulation handled?
Pooled HR means that the values of individual HR provided or extracted from the included studies. Based on the reports of whether the miRNA was up or downregulated, we separated and analysed the effect size in the meta-analysis. Interestingly, the miRNA-200c which was studied in three different studies (Damiano V et al, (2017), Lv J et al. (2014) and Chen J et al. (2011)), only one (Chen J et al (2011)) has reported the HR value.
The introduction gives many epidemiological data on breast cancer, and I do not think that this this really relates to the topic of this study.
Breast cancer is the most prevalent of all the cancers. Therefore, our molecular and clinical oncology team would like to emphasize its importance.
Maybe the selection criteria have been too strict. Is it really necessary that a study needs “both breast cancer patients’ as well as in vitro studies with cell-lines”?
As mentioned in the selection criteria (line no: 151-168), we have included both patients as well as cell-line data wherein the studies providing/ able to extract HR value were very less which is one of the limitations for the meta-analysis.
Many Thanks,
Dr Rama Jayaraj
Meta-Analysis team
Dr Rama Jayaraj PhD, GCTLHE, MPH
Senior Lecturer-Clinical Sciences
College of Health and Human Sciences
+61 8 8946 6146 | F. +61 8 8946 7474
rama.jayaraj@cdu.edu.au
www.cdu.edu.au
CHARLES DARWIN UNIVERSITY
Round 2
Reviewer 3 Report
The authors have submitted a revised version of their manuscript responding to the reviewers ‘comments.
I am not an expert in meta-studies (and many readers will also not having expert knowledge here). Some of my points might have been related to this circumstance and I would like to have a few points clarified for the non-experts.
In detail, I still have problems with the use of the pooled HR-value, representing different types of regulation and chemotherapeutic drugs. Especially, when people are reading the abstract, they see this pooled HR with a confidence interval that is not statistically significant. Nevertheless, this pooled HR is clarified further in the Forrest plot. Here, at least two miRNAs show statistical significance. So I think it would be better to also show such “positive” data in the abstract.
I also have some understanding problems and think other readers might have similar difficulties. So, I suggest that the authors might clarify some points.
Especially concerning the up- and downregulation, I think the authors should make this point clearer. As far as I understood the paper, upregulation refers to cell culture experiments where treated cultures have been compared with untreated or resistant cultures. In a second step the authors extracted an HR for the observed type of miRNA regulation, so i.e. if a miRNA was downregulated in an in-vitro experiment by the drug, the HR was obtained for the cases with low abundance of this miRNA BEFORE treatment.
For two miRNAs, it turned out that although an upregulation was found (in vitro), the HR was >1 (unfavorable) for high abundance. Vice versa, one would expect that a low abundance of this miRNA has a favorable HR. So, this miRNA is still a prognostic biomarker but not as suggested by the up/downregulation. Thus the interpretation of the observed type of regulation could be wrong in the context of a heterogeneous disease.
Author Response
To
Ms Lena Ilic
Assistant Editor
Cells (MDPI)
Special issue: MicroRNA as Biomarker
Ms Lena Ilic
We would like to thank once again the Cells (MDPI), Editorial and Reviewer’s team for reviewing our manuscript and providing valuable comments. Our team firmly believe that the comments and feedback from your esteemed reviewers were constructive and would enhance the quality of the paper. Please find the revised manuscript to address issues identified by editors and reviewers team in considering the above review paper for publication.
We have amended the changes and highlighted the sections in the manuscript with track changes.
Reviewer 3:
I am not an expert in meta-studies (and many readers will also not having expert knowledge here). Some of my points might have been related to this circumstance and I would like to have a few points clarified for the non-experts.
In detail, I still have problems with the use of the pooled HR-value, representing different types of regulation and chemotherapeutic drugs. Especially, when people are reading the abstract, they see this pooled HR with a confidence interval that is not statistically significant. Nevertheless, this pooled HR is clarified further in the Forrest plot. Here, at least two miRNAs show statistical significance. So I think it would be better to also show such “positive” data in the abstract.
We would like to thank the reviewer for their valuable insight.
We have amended two miRNAs show statistical significance as per Reviewer’s recommendation.
We have used the word “combined HR” instead of “pooled HR”, to enable clarity for the readers, as the essential part of meta-analysis is combining the results of multiple studies to generate a combined overall result. We have mentioned about the significant HR values in the abstract (Line no: 42). Furthermore, the core concept behind conducting a meta-analysis is providing an unbiased, impartial analysis, regardless of whether it is “positive” or “negative”, as without doing so, we incur bias which invalidates the entire study. True, there are a few miRNA that show a “positive” and significant result, in terms of their effect, however, these results are not our own. These positive results are the results presented by the individual studies included in this systematic review and meta-analysis.
I also have some understanding problems and think other readers might have similar difficulties. So, I suggest that the authors might clarify some points.
Especially concerning the up- and downregulation, I think the authors should make this point clearer. As far as I understood the paper, upregulation refers to cell culture experiments where treated cultures have been compared with untreated or resistant cultures. In a second step the authors extracted an HR for the observed type of miRNA regulation, so i.e. if a miRNA was downregulated in an in-vitro experiment by the drug, the HR was obtained for the cases with low abundance of this miRNA BEFORE treatment.
We would like to specify that HR, or Hazard Ratio is an end-point based parameter, traditionally used for survival analysis. The values calculated for up-regulation and down-regulation are for the total period of the analysis up until the end-point of the in-vitro analysis. Therefore, the calculation of the HR value encompasses the entire in-vitro experiment, and is not a cross-sectional analysis. We would also like to note that in our study, the up-regulation and down-regulation of miRNA, simply refers to the overexpression and underexpression of the gene that produces said miRNA based on the reference internal control housekeeping genes such as GAPDH and RNU6B. HR cannot be obtained before, after or for any intermediate point in the study (including treatment), it can only be calculated at the end-point.
For two miRNAs, it turned out that although an upregulation was found (in vitro), the HR was >1 (unfavorable) for high abundance. Vice versa, one would expect that a low abundance of this miRNA has a favorable HR. So, this miRNA is still a prognostic biomarker but not as suggested by the up/downregulation. Thus the interpretation of the observed type of regulation could be wrong in the context of a heterogeneous disease.
We agree with the author. However, as our data is synthesized from other previous studies and all conclusions we draw are limited by the fact (it is an inherent limitation of meta-analyses), this will be the case regardless of the depth our analysis, and will persist as a limitation. Therefore we have added the statement that, the possibility of our interpretation being wrong in the context of heterogeneous disease as a limitation of the study.
